# UNCERTAINTY AS FEATURE GAPS:
# EPISTEMIC UNCERTAINTY QUANTIFICATION OF LLMS IN CONTEXTUAL QUESTION-ANSWERING

**Yavuz Bakman**[1]* **Sungmin Kang**[1] **Zhiqi Huang**[2] **Duygu Nur Yaldiz**[1]

**Catarina G. Belém**[3] **Chenyang Zhu**[2] **Anoop Kumar**[2] **Alfy Samuel**[2]

**Salman Avestimehr**[1] **Daben Liu**[2] **Sai Praneeth Karimireddy**[1]

[1]University of Southern California    [2]Capital One    [3]University of California, Irvine

ybakman@usc.edu

## ABSTRACT

Uncertainty Quantification (UQ) research has primarily focused on closed-book factual question answering (QA), while contextual QA remains unexplored, despite its importance in real-world applications. In this work, we focus on UQ for the contextual QA task and propose a theoretically grounded approach to quantify *epistemic uncertainty*. We begin by introducing a task-agnostic, token-level uncertainty measure defined as the cross-entropy between the predictive distribution of the given model and the unknown true distribution. By decomposing this measure, we isolate the epistemic component and approximate the true distribution by a perfectly prompted, idealized model. We then derive an upper bound for epistemic uncertainty and show that it can be interpreted as semantic feature gaps in the given model's hidden representations relative to the ideal model. We further apply this generic framework to the contextual QA task and hypothesize that three features approximate this gap: *context-reliance* (using the provided context rather than parametric knowledge), *context comprehension* (extracting relevant information from context), and *honesty* (avoiding intentional lies). Using a top-down interpretability approach, we extract these features by using only a small number of labeled samples and ensemble them to form a robust uncertainty score. Experiments on multiple QA benchmarks in both in-distribution and out-of-distribution settings show that our method substantially outperforms state-of-the-art unsupervised (sampling-free and sampling-based) and supervised UQ methods, achieving up to a 13-point PRR improvement while incurring a negligible inference overhead. The code is available at https://github.com/Ybakman/Feature-Gaps.

## 1 INTRODUCTION

Despite their impressive performance across a wide range of real-world tasks, Large Language Models (LLMs) still suffer from hallucinations and incorrect generations, which limit their deployment in high-stakes domains such as medicine and finance (Bengio et al., 2025; Ravi et al., 2024). Uncertainty Quantification (UQ) has emerged as a key tool for detecting such errors by only using the model itself, such as output consistency, log-probabilities, or internal activations. Recent works (Bakman et al., 2025; Vashurin et al., 2025) have demonstrated that UQ methods exhibit strong empirical performance across a variety of evaluation benchmarks.

While these results are encouraging, most existing works (Yaldiz et al., 2025a; Lin et al., 2024; Kuhn et al., 2023; Duan et al., 2024) design and evaluate their methods primarily on closed-book factual question answering (QA) tasks, which test the success of UQ methods on the model's memory abilities. Although this direction is important, another critical ability of LLMs that deserves more attention from a UQ perspective is their *contextual* capabilities. With the increasing popularity

---

*Corresponding author & Work done during the internship at Capital One

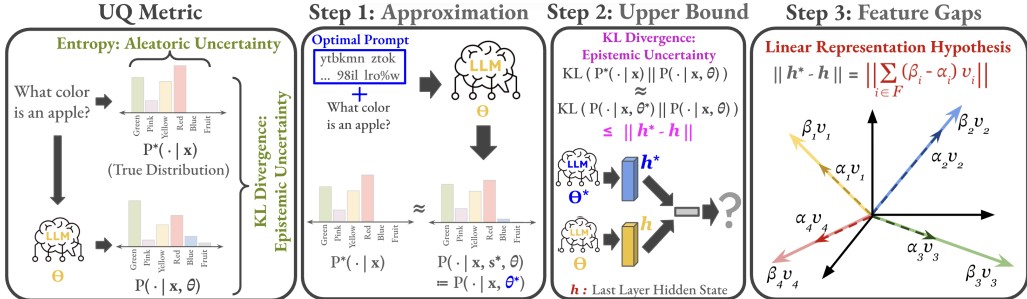

Figure 1: Derivation steps for epistemic uncertainty as feature gaps. Visualization of Section 3.

of Retrieval-Augmented Generation (RAG) in many LLM applications, detecting errors in model generations conditioned on retrieved context has become more important. However, relatively little effort has been devoted to this direction (Soudani et al., 2025; Fadeeva et al., 2025; Perez-Beltrachini & Lapata, 2025), where the proposed approaches often rely on heuristics rather than grounded theory.

Motivated by these observations, we focus on developing a theoretically grounded UQ method for contextual QA. In this setting, the relevant context is either already retrieved or directly provided by the user, and a context-relevant question is posed to the model. Our goal is to quantify the model's uncertainty for a given input as a means to assess whether its output is likely to be correct/reliable.

To quantify the uncertainty of an LLM, we first propose an uncertainty metric defined as the cross-entropy between the true predictive distribution and the given model's distribution, inspired by Schweighofer et al. (2024). Our approach introduces a key modification to their formulation by reversing the position of the true and given model distributions within the cross-entropy, and adapting it specifically for LLMs. We further decompose the total uncertainty into two components: *epistemic* and *aleatoric* uncertainty. In our problem setup, epistemic uncertainty, the model's lack of ability or knowledge to correctly and reliably answer a given question–context pair, is our main interest. After approximating the true predictive distribution with a perfectly prompted hypothetical ideal model, we show that epistemic uncertainty can be bounded by the distance between the last layer hidden state of the given model and the ideal hypothetical model. We further show that this distance can be expressed as the sum of distances over linearly independent model features. Importantly, this result generalizes to any LLM task and is visualized in Figure 1. To approximate this distance specifically in contextual question answering task, we hypothesize three desirable features that capture how far the given model is from the ideal model: 1) **Context reliancy**: the model should ground its answer in the provided context rather than relying on its parametric knowledge; 2) **Context comprehension**: the model should be able to extract and integrate relevant information from the context to answer the question accurately. 3) **Honesty**: the model should avoid intentionally outputting a wrong answer.

Following a top-down interpretability approach similar to Zou et al. (2025), we extract the aforementioned high-level semantic features using a small set of labeled samples to identify the optimal layer for feature extraction. At test time, we combine the activation amount of three features to quantify epistemic uncertainty by computing only three dot products between the model's hidden state and the corresponding feature vectors, one per feature, without requiring any sampling. Our method is highly efficient and achieves substantial performance gains: it outperforms SOTA unsupervised, sampling-free, and sampling-based approaches by up to 16 PRR points. Furthermore, with the same amount of labeled data, it surpasses strong supervised baselines such as SAPLMA (Azaria & Mitchell, 2023) and LookbackLens (Chuang et al., 2024) by up to 13 PRR points, while exhibiting significantly better out-of-distribution generalization compared to SAPLMA, which is an important and desirable property for supervised UQ methods. The overview of our proposed method is visualized in Figure 1.

## 2 PRELIMINARIES

### 2.1 ALEATORIC AND EPISTEMIC UNCERTAINTY

The total uncertainty of a model is typically decomposed into two components: *epistemic* and *aleatoric* uncertainty (Hüllermeier & Waegeman, 2021). Epistemic uncertainty arises from a lack of knowledge. In the context of LLMs, when faced with a difficult question that the model does not know the answer to, its output distribution tends to be more uniform, which indicates uncertainty about which answer is correct. This uncertainty stems from the model's inability or lack of knowledge

to provide the correct answer, and is therefore classified as *epistemic*. In contrast, *aleatoric* (or data) uncertainty captures variability inherent to the task or data, rather than limitations in the model's knowledge. For example, a model may be epistemically confident, knowing the answer, but still produce multiple valid responses due to ambiguity in the question or the presence of multiple equally correct phrasings. This variability arises from the nature of the query and the language itself, not from the model's lack of ability. In the next section, we discuss how existing works conceptualize UQ in LLMs, and how these concepts relate to epistemic and aleatoric uncertainty.

## 2.2 Uncertainty Quantification of LLMs

In the LLM literature, uncertainty quantification is typically used to identify incorrect or unreliable answers generated for a given query. Unlike the well-established frameworks in classification tasks (Gal & Ghahramani, 2016), there is no widely accepted UQ framework for generative LLMs (Bakman et al., 2025; Vashurin et al., 2025). With a few exceptions, most approaches rely on heuristic-based methods that estimate the correctness of a model's (greedy or sampled) generation. UQ methods only use the model itself to find such a score by using signals such as token probabilities (Farquhar et al., 2024), internal representations (Chen et al., 2024), or output consistencies (Lin et al., 2024). Although rarely stated explicitly, the underlying objective in many of these works is to better quantify *epistemic* uncertainty, i.e., to produce an uncertainty score that quantifies the model's (lack of) certainty in the correctness of its own generation. The performance of UQ methods is typically evaluated using threshold-free metrics such as the Area Under the ROC Curve (AUROC) and the Prediction–Rejection Ratio (PRR) to measure which assess how well uncertainty scores distinguish between correct and incorrect outputs. A smaller number of studies (Abbasi-Yadkori et al., 2024; Aichberger et al., 2024) take a more theoretically grounded approach, explicitly distinguishing between epistemic and aleatoric uncertainty. In this work, we also aim to separate epistemic and aleatoric uncertainty through our proposed UQ formulation. In the following section, we introduce our notation and describe the problem setup.

## 2.3 Problem Setup and Notation

We denote the context sequence as $\mathbf{c}$, and the question together with any relevant instructions as $\mathbf{x}$. The probability distribution over the token at position $t$ produced by the model, conditioned on the context $\mathbf{c}$, query $\mathbf{x}$, and previously generated tokens, is given by: $P(y_t \mid \mathbf{y}_{<t}, \mathbf{x}, \mathbf{c}, \theta)$, where $\mathbf{y}_{<t}$ denotes the sequence of tokens generated before timestep $t$, and $\theta$ represents the given model parameters. Our objective is to find an uncertainty quantification method $U(\mathbf{x}, \mathbf{c}, \mathbf{y}) \in \mathbb{R}$ that is *negatively correlated* with the correctness of the generated sequence $\mathbf{y}$. More formally, we aim to maximize $\mathbb{E}\left[\mathbb{1}_{U(\mathbf{x}_1, \mathbf{c}_1, \mathbf{y}_1) < U(\mathbf{x}_2, \mathbf{c}_2, \mathbf{y}_2)} \cdot \mathbb{1}_{\mathbf{y}_1 \in Y_1 \,\wedge\, \mathbf{y}_2 \notin Y_2}\right]$, where $(\mathbf{x}_1, \mathbf{y}_1), (\mathbf{x}_2, \mathbf{y}_2) \sim D_{\text{test}}$, with $D_{\text{test}}$ denoting the evaluation dataset obtained by getting the most probable (greedy) output for a context-query pair, and $Y_i$ representing the set of acceptable (correct) generations for instance $i$. This expectation enforces a ranking consistency: correct outputs should receive lower uncertainty scores than incorrect outputs, making high-uncertainty scored generations more likely to be wrong.

## 3 Bounding Epistemic Uncertainty via Feature Gaps

### 3.1 Uncertainty Quantification Metric

Before introducing our uncertainty quantification metric for LLMs, we define the notion of a *true* (but unknown) token generation distribution, denoted by $P^*(\cdot \mid \mathbf{x})$. This distribution represents the behavior of an ideal model that is free from epistemic uncertainty, i.e., uncertainty arising from incomplete knowledge due to limited data, suboptimal architecture choices, imperfect training, or insufficient instruction tuning. The concept of such epistemically optimal distributions has also been explored in recent works (Kotelevskii et al., 2025; Abbasi-Yadkori et al., 2024).

Given the true distribution $P^*(\cdot \mid \mathbf{x})$ and given model's conditional distribution $P(\cdot \mid \mathbf{x}, \theta)$, we define the *total uncertainty* of a token $y_t$ at generation step $t$ as follows:

**Definition 1** (Total Uncertainty). *Let $\mathcal{V}$ denote the token vocabulary. The total uncertainty (TU) of the model $\theta$ for generating token $y_t$ at timestep $t$, conditioned on the input $x$, is defined as the*

*cross-entropy between the true distribution and the model's predictive distribution:*

$$\text{TU} = - \sum_{y_t \in \mathcal{V}} P^*(y_t \mid \mathbf{y}_{<t}, \mathbf{x}) \cdot \ln P(y_t \mid \mathbf{y}_{<t}, \mathbf{x}, \theta),$$

*where $\mathbf{y}_{<t}$ denotes the previously sampled tokens up to timestep $t$.*

This definition allows us to decompose total uncertainty into *aleatoric* (data) and *epistemic* uncertainty with an intuitive interpretation. The total uncertainty can be expressed as the sum of two terms:

$$\text{TU} = \underbrace{H\left(P^*(y_t \mid \mathbf{y}_{<t}, \mathbf{x})\right)}_{\text{Aleatoric (Data) Uncertainty}} + \underbrace{\text{KL}\left(P^*(y_t \mid \mathbf{y}_{<t}, \mathbf{x}) \,\|\, P(y_t \mid \mathbf{y}_{<t}, \mathbf{x}, \theta)\right)}_{\text{Epistemic Uncertainty}}, \tag{1}$$

The first term, $H(P^*(y_t \mid \mathbf{y}_{<t}, \mathbf{x}))$, is the entropy of the true distribution, which corresponds to *aleatoric* (data) uncertainty. Since the true distribution $P^*(y \mid \mathbf{x})$ has no epistemic uncertainty, any uncertainty in its predictions must arise from inherent randomness in the language modeling data distribution, $(\mathbf{x}, \mathbf{y}) \sim \mathcal{D}$. The second term, $\text{KL}(P^*(y_t \mid \mathbf{y}_{<t}, \mathbf{x}) \,\|\, P(y_t \mid \mathbf{y}_{<t}, \mathbf{x}, \theta))$, measures the divergence between the true distribution and the predictive distribution of the actual model. This gap captures *epistemic* uncertainty, uncertainty arising from the actual model's lack of knowledge or ability compared to the epistemically optimal distribution. Lastly, Schweighofer et al. (2024) recently proposed a UQ metric for classification tasks that instead swaps the positions of $P(y_t \mid \mathbf{y}_{<t}, \mathbf{x}, \theta)$ and $P^*(y_t \mid \mathbf{y}_{<t}, \mathbf{x})$. We discuss the differences between our proposed formulation and theirs, along with the motivation for our choice, in Appendix A.1.

## 3.2 STEP 1: APPROXIMATING THE TRUE DISTRIBUTION

The true predictive distribution $P^*(\cdot \mid \mathbf{x})$ is unknown and intractable, which makes exact computation of epistemic uncertainty infeasible. Therefore, we approximate $P^*(\cdot \mid \mathbf{x})$ through our given model $\theta$. Specifically, we approximate the ideal model as the actual model that has been *perfectly instructed or prompted* so that its output distribution is as close as possible to $P^*(\cdot \mid \mathbf{x})$. Since appending an instruction or prompt can be theoretically viewed as a form of fine-tuning, as shown by many works (Dherin et al., 2025; Akyurek et al., 2023), this approximation corresponds to obtaining the closest possible distribution to $P^*(\cdot \mid \mathbf{x})$ by training the given model in token space. Lastly, prompting is powerful enough to be Turing-complete: for any computable function, there exists a Transformer and a corresponding prompt that computes it (Qiu et al., 2025).

Formally, we approximate the true distribution $P^*(\cdot \mid \mathbf{x})$ by $P(\cdot \mid \mathbf{x}, \mathbf{s}^*, \theta)$, where $\mathbf{s}^* = (s_1, s_2, \ldots, s_n)$ is the optimal token sequence that minimizes the following objective:

$$\mathbf{s}^* := \arg\min_{\mathbf{s}} \mathbb{E}_{\mathbf{x} \sim \mathcal{D}}\left[\text{KL}\left(P^*(\cdot \mid \mathbf{x}) \,\|\, P(\cdot \mid \mathbf{x}, \mathbf{s}, \theta)\right)\right]. \tag{2}$$

where $\mathcal{D}$ is the data distribution of language modeling task. This objective corresponds to finding the optimal pre-sequence such that the resulting output distribution of the model $\theta$ is as close as possible to the true distribution $P^*(\cdot \mid \mathbf{x})$, in expectation over the data distribution $\mathcal{D}$. We refer to the approximated model $P(\cdot \mid \mathbf{x}, \mathbf{s}^*, \theta) \approx P^*(\cdot \mid \mathbf{x})$ as the *ideal model*. For notational simplicity, we denote it by $P(\cdot \mid \mathbf{x}, \theta^*) := P(\cdot \mid \mathbf{x}, \mathbf{s}^*, \theta)$, since the output distribution of the optimally prompted model can be considered as the behavior of an ideal model $\theta^*$. These two models, $\theta$ and $\theta^*$, share the same architecture and weights, but their activations differ due to differences in prompting.

## 3.3 STEP 2: DERIVING AN UPPER BOUND FOR EPISTEMIC UNCERTAINTY

Finding the optimal sequence $\mathbf{s}^*$ requires an exponential enumeration over all possible token sequences, which is computationally infeasible. However, we can derive an upper bound on epistemic uncertainty in terms of the model's internal representations.

**Lemma 1** (Epistemic Uncertainty Upper Bound). *For any token $y_t$,*

$$\text{KL}(P(y_t \mid \mathbf{y}_{<t}, \mathbf{x}, \theta^*) \,\|\, P(y_t \mid \mathbf{y}_{<t}, \mathbf{x}, \theta)) \leq 2\|W\| \, \|h_t^* - h_t\|,$$

*where $h_t^* \in \mathcal{R}^d$ and $h_t \in \mathcal{R}^d$ are the last-layer hidden states of the ideal and actual models with dimension of $d$, respectively, and $W \in \mathcal{R}^{V \times d}$ is the vocabulary projection matrix at the last layer.*

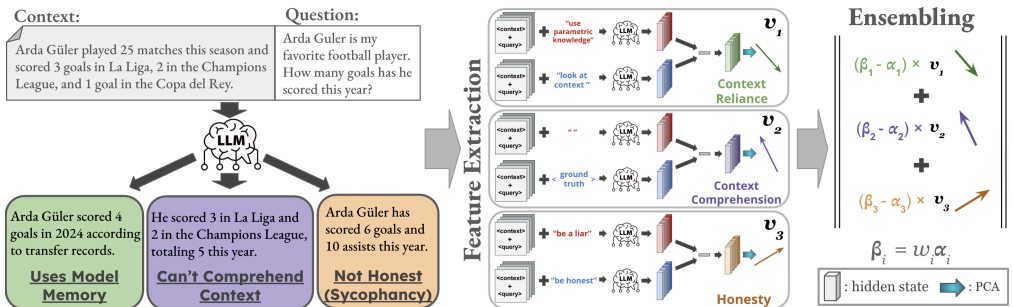

Figure 2: Approximating the bound in contextual QA as an ensemble of three features.

The proof of Lemma 1 begins by expressing the probability distributions in terms of the model's internal representations and leveraging the fact that both models share the same vocabulary projection matrix. The complete derivation is provided in Appendix A.2. Lemma 1 implies that epistemic uncertainty is bounded by the norm of the difference between the last-layer hidden states, scaled by $2\|W\|$. Since $2\|W\|$ is fixed and we are interested in the *relative* magnitude of uncertainty rather than its absolute value, estimating this hidden-state distance is sufficient for our purposes.

### 3.4 STEP 3: INTERPRETING THE UPPER BOUND AS FEATURE GAPS

Although we have bounded epistemic uncertainty in terms of the distance to the last-layer hidden state of the ideal model, the hidden state of $\theta^*$ remains unknown. To better understand this hidden state difference, we leverage one of the key hypotheses in interpretability in language models.

**Hypothesis 1** (Linear Representation (Informal)). *High-level semantic features are encoded approximately linearly in the activation space of language models, often as single directions.*

This hypothesis is broadly accepted and supported by substantial empirical evidence from prior work (Park et al., 2024; Nanda et al., 2023; Templeton et al., 2024). As an example, we can identify a vector in intermediate layers that corresponds to a feature such as "toxicity": the activation along this direction increases when the model produces toxic outputs, and decreases otherwise.

Let $h_t$ and $h_t^*$ denote the $d$-dimensional last layer hidden states of the actual and ideal models, respectively. Due to residual connections, both $h_t$ and $h_t^*$ can be written as the sum of layer outputs: $h_t = \sum_{l=1}^{L} h_t^l$ and $h_t^* = \sum_{l=1}^{L} h_t^{l*}$, where $h_t^l$ is the output of layer $l$ at timestep $t$. For a desired decomposition at layer $l$, let $\mathcal{F}^l$ be a set of feature vectors $v_i^l \in \mathbb{R}^d$, where $|\mathcal{F}^l| \geq d$ and $\text{rank}(\mathcal{F}^l) = d$. Then, the hidden states can be expressed as: $h_t = \sum_{l=1}^{L} \sum_{v_i^l \in \mathcal{F}^l} \alpha_i^l v_i^l$ and $h_t^* = \sum_{l=1}^{L} \sum_{v_i^l \in \mathcal{F}^l} \beta_i^l v_i^l$, where $\alpha_i^l$ and $\beta_i^l$ are the coefficients for the actual and ideal models, respectively. For simplicity, let $\mathcal{F} = \mathcal{F}^1 \cup \mathcal{F}^2 \cup \cdots \cup \mathcal{F}^L$. Then we can write: $h_t = \sum_{v_i \in \mathcal{F}} \alpha_i v_i$ and $h_t^* = \sum_{v_i \in \mathcal{F}} \beta_i v_i$.

The norm of the difference between the hidden states then becomes:

$$\|h_t^* - h_t\| = \left\| \sum_{v_i \in \mathcal{F}} (\beta_i - \alpha_i) v_i \right\|. \tag{3}$$

Note that the above derivation holds for any language model with the same architecture without the linear representation hypothesis. However, the linear representation hypothesis provides a crucial interpretability advantage: it allows us to decompose each layer into semantically meaningful features $\mathcal{F}$. Since the actual model and the ideal model differ only by the input prompt, sharing the same architecture and weights, their feature vectors $v_i$ correspond to the same semantic concepts. This alignment enables us to interpret the error term in Equation 3 as a collection of *feature gaps*, $(\beta_i - \alpha_i)$, that quantify how the actual model deviates from the ideal model along interpretable semantic directions. This whole introduced UQ framework is visualized in Figure 1

**Remark 1.** *All derivations up to this point have been generic to any language modeling task, such as factual QA, mathematics, coding, or contextual QA.*

# 4 COMPUTING EPISTEMIC UNCERTAINTY IN CONTEXTUAL QA

## 4.1 SELECTING A REPRESENTATIVE FEATURE SET FOR CONTEXTUAL QA

Since the error term in Equation 3 may consist of arbitrarily many feature directions $v_i$, computing the bound exactly is infeasible. We therefore hypothesize a small set of features that are most likely to contribute to the gap between the actual and ideal models in the contextual QA setting. For example, selecting a syntactic feature would not be meaningful, as modern LLMs already exhibit strong mastery of grammar and sentence structure (OpenAI, 2023). Thus, the potential gap along such dimensions is expected to be negligible. Instead, we focus on the features where current LLMs are more likely to deviate from the ideal model. Intuitively, if the model grounds its answer in the provided context, fully comprehends the contextual information, and outputs its understanding honestly, then it can behave similarly to the ideal model. Following this reasoning, we hypothesize that three high-level semantic features can approximate the gap in Equation 3: The first is **Context Reliance**: in contextual QA, the model's parametric knowledge may contradict the information contained in the provided context (Longpre et al., 2021). Models often default to their internal knowledge (which may be outdated or incorrect), resulting in unreliable answers. The second is **Context Comprehension**: in many contextual QA tasks, the answer may not be explicitly stated in the context but must be extracted or inferred from it. The third is **Honesty**: language models may sometimes generate false information deliberately. An example of this is *LLM sycophancy* (Sharma et al., 2025), where the model fabricates answers to align with user expectations rather than admitting ignorance. Now, our approximation becomes: $\left\| \sum_{v_i \in \mathcal{F}} (\beta_i - \alpha_i) v_i \right\| \approx \left\| \sum_{v_i \in \mathcal{H}} (\beta_i - \alpha_i) v_i \right\|$ where $\mathcal{F}$ denotes the full set of latent features and $\mathcal{H}$ is the restricted set consisting of the three features defined above.

## 4.2 FEATURE EXTRACTION AND ENSEMBLING

**Feature Extraction.** To extract these three features, we adopt a top-down interpretability approach similar to Zou et al. (2025), which requires only a small amount of labeled data. Suppose we have access to a set of $T$ labeled samples, each consisting of a question–instruction pair $\mathbf{x}$ and a context $\mathbf{c}$. We first obtain the greedy answer $\mathbf{y}$ from the model under the standard instruction. For each feature, we then construct contrastive instruction–input pairs designed to isolate that feature. Concretely, this involves two forward passes with carefully designed prompts. For example, to capture the **context reliance** feature, we run one forward pass with the instruction "look at the context" and another with "use your own knowledge". This difference is expected to capture the "context-parametric knowledge reliance" direction in representation space. After repeating this procedure over the dataset, we find the strongest direction through PCA, which corresponds to the desired feature vector. Formally:

$$m_i^l = \theta_l(\mathbf{y}_i, \mathbf{x}_i + \text{"look at the context"}, \mathbf{c}_i) - \theta_l(\mathbf{y}_i, \mathbf{x}_i + \text{"use your own knowledge"}, \mathbf{c}_i), \quad (4)$$

$$M^l = [m_1^l, m_2^l, \ldots, m_T^l], \quad (5)$$

$$v^l = \text{PCA}(M^l) \quad (6)$$

where $\theta_l$ denotes the hidden representation at layer $l$. We follow a similar procedure for the other two features. For **context comprehension**, we perform one pass with the original context $\mathbf{c}$ and another with $\mathbf{c} + \text{"{ ground truth }"}$. This ground truth append simulates the model having already resolved the relevant information from the context, thereby isolating the context comprehension feature. For **honesty**, we contrast the instructions "be honest" versus "be a liar."

**Selecting Optimal Layers and Ensembling.** We use the same dataset employed for feature extraction to select the most informative layer for each feature. For each sample and each layer $l$, we compute the dot product between the hidden state $h_l$ (averaged across all tokens in $\mathbf{y}$) and the extracted feature vector $v^l$: $s^l = h^{l\top} v^l$. We then measure the correlation between these scores $[s_1^l, s_2^l, \ldots, s_T^l]$ and the generation correctness using PRR. The layer with the highest PRR is selected as the feature layer.

To ensemble the three features, we need to estimate the coefficients $\beta_i$. In principle, any function could be used for this estimation. For simplicity, we define $\beta_i = w_i \alpha_i$, i.e., a scaled version of $\alpha_i$, where the scaling factors $w_i$ are trained to minimize the cross-entropy error with respect to the correctness of a generation. This formulation reduces the learning problem to training only three parameters $(w_1, w_2, w_3)$, which is why only a small number of labeled samples is sufficient. After

learning the weights, the final ensemble reduces to a linear combination of all three features:

$$\sum_{v_i \in \mathcal{H}} (\beta_i - \alpha_i) v_i = \sum_{v_i \in \mathcal{H}} (w_i - 1)\alpha_i v_i. \tag{7}$$

## 5 EXPERIMENTS

### 5.1 EXPERIMENTAL SETUP

**Datasets.** We evaluate our approach on three contextual question answering datasets: (i) **Qasper** (Dasigi et al., 2021), a dataset for question answering over scientific research papers; (ii) **HotpotQA** (Yang et al., 2018), a Wikipedia-based dataset consisting of multi-hop question–answer pairs with supporting passages provided; and (iii) **NarrativeQA** (Kočiský et al., 2018), a dataset of stories and associated questions designed to test reading comprehension, particularly over long documents. We use 1000 samples from the each dataset.

**Models.** We use three models: `LLaMA-3.1-8B`, `Mistral-v0.3-7B`, and `Qwen2.5-7b`.

**Performance Metrics.** We evaluate uncertainty quantification methods using two widely adopted metrics (Vashurin et al., 2025): Area Under the Receiver Operating Characteristic Curve (AUROC) and Prediction–Rejection Ratio (PRR). AUROC measures a method's ability to discriminate between correct and incorrect outputs across all possible thresholds, with values ranging from 0.5 (random performance) to 1.0 (perfect discrimination). PRR quantifies the relative precision gain achieved by rejecting low-confidence predictions, ranging from 0.0 (random rejection) to 1.0 (perfect rejection).

**Correctness Measure.** As our tasks involve free-form generation, model outputs may be semantically correct even when they do not exactly match the reference answers lexically. To account for this, we adopt the *LLM-as-a-judge* paradigm, following prior work (Bakman et al., 2025; Farquhar et al., 2024). Concretely, we prompt a language model (`Gemini-2.5-flash`) with the question, the generated answer, the reference answer, and the context, and ask it to output a correctness judgment.

**Baselines.** We compare our method against several widely used unsupervised and supervised UQ methods by using TruthTorchLM library (Yaldiz et al., 2025b). Specifically, we include: **Perplexity** (Malinin & Gales, 2021), which computes the average negative log-probability of the greedy output; **Entropy** (Malinin & Gales, 2021), which samples multiple generations and averages their log-probabilities; **Semantic Entropy** (Farquhar et al., 2024), which samples generations, clusters semantically equivalent outputs, and then computes entropy over the clusters; **MARS** (Bakman et al., 2024), which weights token probabilities by their contribution to meaning; **SAR** (Duan et al., 2024), which incorporates relevance scores of sampled generations into the entropy calculation and weights tokens by their relevancy to the sentence; **Mini-Check** (Tang et al., 2024), which trains a small model to check logical entailment between the generation and the context;**LLM-Judge** (Zheng et al., 2023), which queries an LLM directly to verify whether a generation is supported by the provided context; **PTrue** Kadavath et al. (2022a) which asks the model's generation corretness and get the probability of token "True" at the end; **Kernel Language Entropy (KLE)** (Nikitin et al., 2024) and **Eccentricity** (Lin et al., 2024), both of which sample multiple generations, compute pairwise similarities, and apply linear-algebraic operations to quantify uncertainty; **SAPLMA** (Azaria & Mitchell, 2023), a supervised approach that trains a classifier on the internal hidden states of the model to predict correctness; **Average Token-level Mahalanobis Distances (ATMD)** Vazhentsev et al. (2025), which calculates the Mahalanobis distance between generated tokens and the average of correct output tokens in the training set, then train a classifier which takes distances as input and predict the correctnes of the generation; **LookBackLens** (Chuang et al., 2024), another supervised method that leverages attention ratios between generated tokens and context tokens. For all methods requiring sampling, we generate 5 samples per input. For all supervised methods, we use a total of 256 labeled examples. Additional experiments in lower data regimes (64 and 128 labeled samples) are presented in Section 5.5.

### 5.2 RESULTS

The results of our method compared to the baselines are presented in Table 1. Our approach achieves consistently superior performance (first or second rank) in both PRR and AUROC across all datasets and models, with the sole exception of `Mistral-7B` on NarrativeQA. We attribute this drop in performance to the limited context window of `Mistral-7B` (32k tokens) relative to the long

contexts in NarrativeQA (13.3% of samples exceed 32k tokens). As a result, the model may fail to produce reliable feature activations for such long contexts, which lie outside its effective training distribution (potentially even shorter than the theoretical 32k limit (Hsieh et al., 2024)). Moreover, our method requires neither sampling nor additional forward passes, which makes it substantially faster than sampling-based approaches such as Semantic Entropy, KLE, and Eccentricity. Lastly, LookBackLens could only be evaluated on the HotpotQA dataset. For the other datasets (Qasper and NarrativeQA), extracting all attention weights was computationally infeasible with the HuggingFace implementation/interface on $8 \times 40$GB NVIDIA A100 GPUs, as it resulted in out-of-memory errors.

| Model | Category | UQ Method | Qasper | | HotpotQA | | NarrativeQA | |
|---|---|---|---|---|---|---|---|---|
| | | | PRR | AUROC | PRR | AUROC | PRR | AUROC |
| **LLama3.1 - 8B** | **No Sampling** **Unsupervised** | **Perplexity** | 47.7 | 68.8 | 50.8 | 69.9 | 57.9 | 72.6 |
| | | **MARS** | 48.3 | 68.4 | 46.6 | 67.9 | 56.4 | 72.2 |
| | | **MiniCheck** | 48.5 | 68.2 | 26.7 | 61.9 | 24.1 | 59.3 |
| | | **LLM-Judge** | 35.7 | 60.7 | 12.1 | 55.6 | 15.4 | 54.1 |
| | | **Ptrue** | 57.4 | 74.2 | 46.1 | 68.8 | 36.7 | 68.2 |
| | **Multi-Sampling** **Unsupervised** | **Entropy** | 29.1 | 58.4 | 41.0 | 63.1 | 39.7 | 62.1 |
| | | **KLE** | 43.9 | 66.4 | 39.8 | 68.7 | 47.3 | 71.6 |
| | | **Eccentricity** | 42.1 | 66.1 | 42.7 | 70.0 | 50.0 | 73.2 |
| | | **SAR** | 53.9 | 71.9 | 53.5 | 71.7 | **59.7** | **75.2** |
| | | **Semantic Entropy** | 42.7 | 67.2 | 47.6 | 69.0 | 51.9 | 72.3 |
| | **No Sampling** **Supervised** | **ATMD** | 32.0 | 62.8 | 26.8 | 60.9 | 21.1 | 57.1 |
| | | **LookBackLens** | - | - | 53.3 | 73.4 | - | - |
| | | **SAPLMA** | 59.9 | 74.7 | 53.0 | 72.8 | 47.3 | 67.5 |
| | | **Feature-Gaps (ours)** | **64.9** | **75.3** | **66.6** | **78.0** | **59.7** | 74.0 |
| **Mistralv0.3 - 7B** | **No Sampling** **Unsupervised** | **Perplexity** | 51.2 | 70.4 | 28.8 | 62.5 | 43.0 | 67.8 |
| | | **MARS** | 54.8 | 73.2 | 25.7 | 60.6 | 47.0 | 69.6 |
| | | **MiniCheck** | 28.3 | 63.2 | 44.0 | 69.4 | 35.9 | 66.1 |
| | | **LLM-Judge** | 39.2 | 65.5 | 28.8 | 64.0 | 22.4 | 61.4 |
| | | **Ptrue** | -51.9 | 36.7 | -9.68 | 49.3 | 9.71 | 56.8 |
| | **Multi-Sampling** **Unsupervised** | **Entropy** | 51.3 | 70.3 | 34.3 | 64.3 | 40.1 | 65.5 |
| | | **KLE** | 33.4 | 63.1 | 45.8 | 71.9 | 48.6 | 74.7 |
| | | **Eccentricity** | 37.7 | 65.2 | 44.1 | 70.7 | **55.5** | **76.3** |
| | | **SAR** | 54.9 | 71.3 | 36.0 | 68.1 | 51.0 | 70.9 |
| | | **Semantic Entropy** | 51.6 | 69.6 | 42.1 | 68.9 | 54.4 | 75.1 |
| | **No Sampling** **Supervised** | **ATMD** | 37.7 | 66.4 | 43.4 | 68.1 | 21.6 | 69.5 |
| | | **LookBackLens** | - | - | 52.2 | 71.4 | - | - |
| | | **SAPLMA** | 44.4 | 69.1 | 53.2 | **73.3** | 53.8 | 71.3 |
| | | **Feature-Gaps (ours)** | **59.7** | **75.9** | **54.2** | 71.4 | 38.5 | 65.1 |
| **Qwen2.5 - 7b** | **No Sampling** **Unsupervised** | **Perplexity** | 42.7 | 66.9 | 27.9 | 58.6 | 42.3 | 65.2 |
| | | **MARS** | 42.4 | 66.1 | 26.5 | 57.8 | 42.7 | 65.0 |
| | | **MiniCheck** | 41.8 | 65.0 | 48.3 | 71.7 | 31.4 | 60.9 |
| | | **LLM-Judge** | 10.9 | 52.7 | 19.6 | 54.5 | 8.6 | 50.7 |
| | | **Ptrue** | 37.4 | 63.9 | -9.68 | 49.3 | 7.79 | 54.1 |
| | **Multi-Sampling** **Unsupervised** | **Entropy** | 41.9 | 67.0 | 28.8 | 59.5 | 43.4 | 66.0 |
| | | **KLE** | 34.5 | 62.8 | 29.4 | 66.9 | 45.5 | 69.9 |
| | | **Eccentricity** | 28.5 | 64.1 | 32.2 | 67.5 | 42.3 | 70.1 |
| | | **SAR** | 45.1' | 67.4 | 36.5 | 65.6 | 48.5 | 69.0 |
| | | **Semantic Entropy** | 41.4 | 67.1 | 35.8 | 65.1 | 47.5 | 69.5 |
| | **No Sampling** **Supervised** | **ATMD** | 37.7 | 66.4 | 43.4 | 68.1 | 21.6 | 59.5 |
| | | **LookBackLens** | - | - | 60.0 | 74.5 | - | - |
| | | **SAPLMA** | **59.1** | **75.2** | 57.9 | **76.2** | 45.1 | 68.8 |
| | | **Feature-Gaps (ours)** | 58.5 | 73.3 | **62.6** | 76.1 | **51.0** | **70.7** |

Table 1: AUROC and PRR performances of UQ methods on Qasper, HotpotQA, and NarrativeQA.

## 5.3 OUT-OF-DISTRIBUTION EVALUATION

A key challenge for supervised UQ methods is their performance under distribution shift, i.e., when the test distribution differs from the training data. To evaluate robustness, we evaluate two supervised methods, our method and SAPLMA, under out-of-distribution (OOD) settings. For each of the three datasets, we construct a $3 \times 3$ train-test matrix, where we train on one dataset in a pair and test on the other. The results, shown in Figure 3, demonstrate that our method is more robust to distribution shifts compared to SAPLMA. This indicates that our feature-based formulation generalizes more effectively across domains, which provides more reliable uncertainty estimates compared to direct supervised training on model activations.

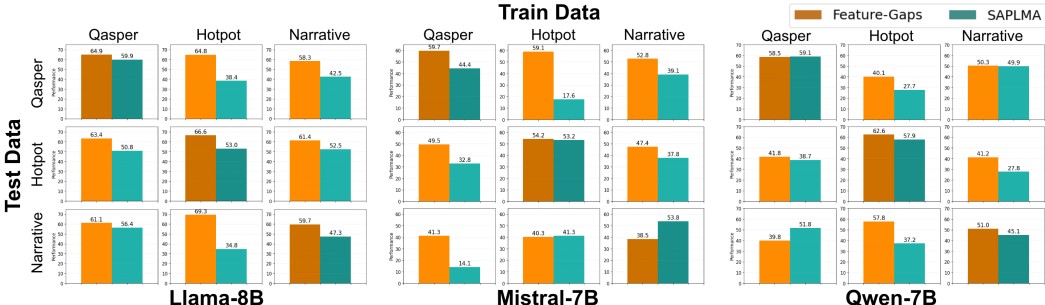

Figure 3: Out-of-distribution evaluation results. In-distribution performances are with darker shades.

## 5.4 PERFORMANCE OF INDIVIDUAL FEATURES

An important ablation study is to understand how much of the performance gain comes from the ensemble itself, compared to the contribution of individual features. To investigate this, we evaluate each feature separately across all model–dataset pairs, measuring PRR as an indicator of its ability to predict correctness (i.e., to serve as a reliable measure of epistemic uncertainty). Results are reported in Table 2. We find that individual features already act as strong epistemic uncertainty estimators on their own. The ensemble offers little to no additional performance gain in terms of PRR. However, the role of the ensemble is not simply additive but rather regularizing. The best-performing feature varies depending on the dataset and model because of the inherent randomness of the proposal, which uses a small number of labeled examples for feature extraction. As shown in Table 2, the top feature differs across datasets. In such cases, the ensemble balances these fluctuations, which yields a more stable and consistent uncertainty quantifier across datasets and, importantly, under OOD conditions (see Section 5.3).

|  | Features | Qasper | HotpotQA | NarrtvQA |
|---|---|---|---|---|
| **LLama** | **Honesty** | 62.0 | 57.7 | 56.7 |
|  | **C. Rel.** | 43.6 | 38.8 | -16.9 |
|  | **C. Comp.** | 59.6 | 66.8 | 52.2 |
|  | **Ensemble** | 64.9 | 66.6 | 59.7 |
| **Mistral** | **Honesty** | 51.4 | 54.9 | 37.3 |
|  | **C. Rel.** | 60.7 | 52.4 | 21.8 |
|  | **C. Comp.** | 27.4 | 52.3 | 21.4 |
|  | **Ensemble** | 59.6 | 54.2 | 38.5 |
| **Qwen** | **Honesty** | 52.9 | 35.1 | 44.2 |
|  | **C. Rel.** | 42.5 | 56.9 | 48.0 |
|  | **C. Comp.** | 33.5 | 61.8 | 56.9 |
|  | **Ensemble** | 58.5 | 62.6 | 51.0 |

Table 2: PRR scores of individual features on Qasper, HotpotQA, and NarrativeQA.

## 5.5 PERFORMANCE IN LOW DATA REGIMES

All supervised methods, including ours, are primarily evaluated using 256 labeled samples. However, the performance of our approach under more limited supervision is critical for its applicability in extreme low-data settings. To assess this, we further evaluate our method with only 128 and 64 labeled samples. Results are reported in Table 3. The findings are encouraging: with 128 samples, performance is largely preserved, showing only marginal degradation compared to the 256-sample setting. Even with as few as 64 samples, although some performance drop is observed, our method remains substantially stronger than alternative baselines reported in Table 1. These results demonstrate that our approach is highly data-efficient and remains effective even in extreme low-data regimes, which highlights its practicality for real-world scenarios where labeled correctness data is scarce.

|  | Num Samples | Qasper | HotpotQA | NarrtvQA |
|---|---|---|---|---|
| **LLama** | **64** | 64.4 | 57.0 | 57.5 |
|  | **128** | 63.2 | 62.0 | 63.2 |
|  | **256** | 64.9 | 66.6 | 59.7 |
| **Mistral** | **64** | 38.3 | 49.5 | 39.4 |
|  | **128** | 52.2 | 55.2 | 38.7 |
|  | **256** | 59.7 | 54.2 | 38.5 |
| **Qwen** | **64** | 41.1 | 60.5 | 37.2 |
|  | **128** | 51.8 | 60.9 | 52.0 |
|  | **256** | 58.5 | 62.6 | 51.0 |

Table 3: PRR performances of Feature-Gaps on low data regimes.

## 5.6 COMPARISON WITH BASELINE DIRECTIONS

Demonstrating the effectiveness of each component of our method is essential for a rigorous scientific evaluation. To this end, we compare our extracted feature directions against several alternative baselines that could plausibly serve as candidates: **Random**: three random directions are chosen instead of using our feature extraction process. **Positive-PCA**: PCA is applied directly on positive samples (e.g. "be honest"), omitting the contrastive difference step. **Negative-PCA**: similar to Positive, but using only negative samples (e.g. "be a liar"). **All-PCA**: the strongest direction is extracted from regular prompts without forming contrastive pairs. **Mean-Diff**: a supervised baseline similar to SAPLMA, where we compute the mean hidden states of correct and incorrect samples at each layer and use their difference as a correctness direction.

The results, shown in Table 4, highlight the

|  | Directions | Qasper | HotpotQA | NarrtvQA |
|---|---|---|---|---|
| LLama | Random | 34.5 | 29.5 | 17.4 |
|  | Positive-PCA | 45.4 | 47.1 | 46.2 |
|  | Negative-PCA | 40.5 | 61.3 | 54.0 |
|  | All-PCA | 4.0 | 26.1 | 18.1 |
|  | Mean-Diff | 48.5 | 53.1 | 36.6 |
|  | Feature-Gaps | 64.9 | 66.6 | 59.7 |
| Mistral | Random | 11.1 | 24.4 | 7.6 |
|  | Positive-PCA | 39.0 | 45.4 | 41.8 |
|  | Negative-PCA | 52.0 | 52.9 | 33.2 |
|  | All-PCA | 4.1 | 36.7 | 12.8 |
|  | Mean-Diff | 51.7 | 49.0 | 48.5 |
|  | Feature-Gaps | 59.6 | 54.2 | 38.5 |
| Qwen | Random | 17.9 | 4.3 | 6.1 |
|  | Positive-PCA | -3.6 | 26.7 | 32.6 |
|  | Negative-PCA | 2.2 | 31.3 | 36.4 |
|  | All-PCA | -4.3 | 20.1 | 36.0 |
|  | Mean-Diff | 57.3 | 49.3 | 47.6 |
|  | Feature-Gaps | 58.5 | 62.6 | 51.0 |

Table 4: PRR scores of baseline directions on Qasper, HotpotQA, and NarrativeQA.

importance of our design choices. Ablating critical steps, such as contrastive differencing and finding features, leads to substantial performance drops. Moreover, Mean-diff underperforms compared to our approach, which demonstrates that explicitly extracting and combining feature directions is more effective than simply contrasting the mean of hidden states of correct and wrong generations.

## 6 CONCLUSION

In this work, we introduced a task-agnostic metric for total uncertainty. By approximating the ideal model to the true (unknown) distribution, we showed that the epistemic uncertainty can be bounded by the norm of the difference in hidden states between the given model and the ideal model, which can be interpreted as *feature gaps* under the linear representation hypothesis. We then applied this framework to contextual QA and hypothesized that three features, *context-reliance*, *context comprehension*, and *honesty*, serving as effective approximations of this gap. Using only a small number of labeled samples, our method achieves superior performance compared to popular baselines. We believe this framework provides a foundation for future research on epistemic uncertainty, including the discovery of additional features and the development of automatic, task-agnostic feature extraction methods, ultimately enabling more robust and generalizable epistemic uncertainty quantifiers.

## 7 LIMITATIONS AND FUTURE WORK

Our method currently requires supervised examples. Making this framework work with unlabeled data, or even without any external data by relying solely on synthetically generated samples, would be an important direction for future research. Although our experiments focus on contextual QA, the feature-gaps framework is general and can be applied to a wide range of language-model tasks. Extending it to other domains such as reasoning or long-form generations, is another promising direction. At present, our method relies on heuristic feature selection. However, this is not a fundamental limitation: developing an automatic procedure for selecting features would be a valuable extension. In addition, our experiments use only a small number of labeled samples (256). Scaling the framework to larger training sets is important. This can be achieved by selecting more features to better approximate the gap, or by selecting more layers per feature, as we demonstrated in Appendix A.5.1. Lastly, in our experiments, we don't consider the potential noise coming from the context-retriever/provider. Retrieving the wrong/useless context is common in RAG systems, and that noise/uncertainty can be modeled in future works. Overall, we believe the feature-gaps framework offers a strong approximation to epistemic uncertainty and has the potential to generalize better than other supervised approaches such as SAPLMA.

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

## A APPENDIX / SUPPLEMENTAL MATERIAL

### A.1 COMPARISON WITH THE INFORMATION-THEORETIC UNCERTAINTY QUANTIFIER OF SCHWEIGHOFER ET AL. (2024)

Schweighofer et al. (2024) propose to quantify total uncertainty in classification tasks as

$$\text{TU} = -\sum_{y \in \mathcal{C}} P(y \mid \mathbf{x}, \theta) \cdot \ln P^*(y \mid \mathbf{x}, \theta),$$

where $\mathcal{C}$ is the set of classes. In their formulation, the roles of the actual model $P(y \mid \mathbf{x}, \theta)$ and the true distribution $P^*(y \mid \mathbf{x}, \theta)$ are swapped compared to ours. The natural decomposition of their metric is

$$\text{TU} = \underbrace{H(P(y \mid \mathbf{x}, \theta))}_{\text{Aleatoric (Data) Uncertainty}} + \underbrace{\text{KL}(P(y \mid \mathbf{x}, \theta) \,\|\, P^*(y \mid \mathbf{x}))}_{\text{Epistemic Uncertainty}}.$$

We argue that this decomposition is problematic. Data (aleatoric) uncertainty should arise from the input $\mathbf{x}$ or the data distribution of the task $\mathcal{D}$, and should be independent of the specific training outcome. While the actual model $\theta$ is indeed trained on a sampled set of $\mathcal{D}$, $D_{\text{train}} \sim \mathcal{D}$, the sampled data may be insufficient and may lead to epistemically sub-optimal models. Besides, training a model is not a deterministic function of $D_{\text{train}}$, different random seeds and hyperparameter settings can yield infinitely many possible models, $\theta_{\text{random}} \xrightarrow{D_{\text{train}}} \theta \sim \Theta$. Consequently, properties of $\theta$ cannot not directly determine the data uncertainty.

Consider an extreme case: if we train $\theta$ with pathological hyperparameters (e.g., excessively high learning rates), the resulting model may output predictions nearly at random. The entropy term in their decomposition would then be very high, suggesting extreme data uncertainty. Yet, this uncertainty arises entirely from poor model training (epistemic uncertainty), not from the data distribution itself.

By contrast, in our formulation where $P(y \mid \mathbf{x}, \theta)$ and $P^*(y \mid \mathbf{x}, \theta)$ are swapped, the aleatoric component is defined in terms of $P^*(y \mid \mathbf{x}, \theta)$, which is independent of any part of training where epistemic uncertainty could arise. Lastly, a more recent work Kotelevskii et al. (2025) also does a similar formulation with ours from the Bayesian risk perspective (see Table 1). For these reasons, we argue that our quantifier provides a more reasonable decomposition of epistemic and aleatoric uncertainty. Nonetheless, we acknowledge that the formulation of Schweighofer et al. (2024) was an important inspiration for our work and served as a foundation for adapting these ideas to language models.

## A.2 Proof of Lemma 1

**Proof.** For notational simplicity, let us denote
$$P(y_t \mid \theta^*) = P(y_t \mid \mathbf{y}_{<t}, \mathbf{x}, \theta^*).$$
We begin by explicitly writing the KL term
$$\mathrm{KL}(P(y_t \mid \theta^*) \,\|\, P(y_t \mid \theta))$$
$$\mathrm{KL}(P(y_t \mid \theta^*) \,\|\, P(y_t \mid \theta)) = \sum_{i \in \mathcal{V}} P(y_i \mid \theta^*) \ln \frac{P(y_i \mid \theta^*)}{P(y_i \mid \theta)}.$$

Since the probability of a token under model $\theta$ is given by
$$P(y_i \mid \theta) = \frac{e^{V_i^\top W h_t}}{\sum_{j \in \mathcal{V}} e^{V_j^\top W h_t}},$$
where $W \in \mathbb{R}^{|\mathcal{V}| \times d}$ is the vocabulary projection matrix and $V_i$ is the one-hot vector of token $y_t$ for token $i$, we can re-write KL in terms of model internals:
$$\sum_{i \in \mathcal{V}} P(y_i \mid \theta^*) \cdot V_i^\top W (h_t^* - h_t) + \sum_{i \in \mathcal{V}} P(y_i \mid \theta^*) \left( \ln \sum_{j \in \mathcal{V}} e^{V_j^\top W h_t} - \ln \sum_{j \in \mathcal{V}} e^{V_j^\top W h_t^*} \right).$$

as both models share the same vocabulary matrix $W$. Focusing on the first term, we have
$$\sum_{i \in \mathcal{V}} P(y_i \mid \theta^*) \cdot V_i^\top W (h_t^* - h_t) \leq \sum_{i \in \mathcal{V}} P(y_i \mid \theta^*) \|V_i\| \|W(h_t^* - h_t)\|$$
by Cauchy–Schwarz. Since $V_i$ is a one-hot vector, $\|V_i\| = 1$, so this simplifies to
$$\sum_{i \in \mathcal{V}} P(y_i \mid \theta^*) \cdot \|W(h_t^* - h_t)\| = \|W(h_t^* - h_t)\|,$$
because $\sum_{i \in \mathcal{V}} P(y_i \mid \theta^*) = 1$. Moreover, by Cauchy-Schwarz inequality,
$$\|W(h_t^* - h_t)\| \leq \|W\| \|h_t^* - h_t\|.$$

For the second term, observe that
$$\sum_{i \in \mathcal{V}} P(y_i \mid \theta^*) \left( \ln \sum_{j \in \mathcal{V}} e^{V_j^\top W h_t} - \ln \sum_{j \in \mathcal{V}} e^{V_j^\top W h_t^*} \right) = \ln \sum_{j \in \mathcal{V}} e^{V_j^\top W h_t} - \ln \sum_{j \in \mathcal{V}} e^{V_j^\top W h_t^*},$$
since $\sum_{i \in \mathcal{V}} P(y_i \mid \theta^*) = 1$.

Define $f(x) := \ln\left( \sum_{i=1}^d e^{x_i} \right)$, the log-sum-exp function. Then
$$\ln \sum_{j \in \mathcal{V}} e^{V_j^\top W h_t} - \ln \sum_{j \in \mathcal{V}} e^{V_j^\top W h_t^*} = f(W h_t) - f(W h_t^*).$$

By the mean value theorem, there exists $c$ on the line segment between $W h_t$ and $W h_t^*$ such that
$$f(W h_t) - f(W h_t^*) = \nabla f(c)^\top (W h_t - W h_t^*).$$
Since $\nabla f(x) = \mathrm{softmax}(x)$, we have
$$f(W h_t) - f(W h_t^*) = \mathrm{softmax}(c)^\top (W h_t - W h_t^*) \leq \|\mathrm{softmax}(c)\| \|W h_t - W h_t^*\| \leq \|W h_t - W h_t^*\|,$$
because $\|\mathrm{softmax}(c)\| \leq 1$. Lastly, $\|W h_t - W h_t^*\| \leq \|W\| \|h_t^* - h_t\|$

**Combining both terms.** From the above bounds, we conclude

$$\text{KL}(P(y_t \mid \theta^*) \,\|\, P(y_t \mid \theta)) \leq 2 \,\|W\| \,\|h_t^* - h_t\|.$$

### A.3 RELATED WORK

A large body of recent work has focused on Uncertainty Quantification (UQ) for language models. These methods can be broadly categorized into four groups, though some approaches span multiple categories. Most existing methods are heuristic in nature:

1. **Output-probability based methods**, such as Semantic Entropy (Kuhn et al., 2023), Sequence-Probability (Aichberger et al., 2024), Mutual Information (Abbasi-Yadkori et al., 2024), MARS (Bakman et al., 2024), LARS (Yaldiz et al., 2025a), and SAR (Duan et al., 2024). 2. **Output-consistency based methods**, including Kernel Language Entropy (Nikitin et al., 2024), Eccentricity, and Matrix-Degree (Lin et al., 2024). 3. **Internal-state based methods**, such as INSIDE (Chen et al., 2024) and SAPLMA (Azaria & Mitchell, 2023). 4. **Self-checking methods**, such as Verbalized Confidence (Tian et al., 2023) and PTrue (Kadavath et al., 2022b).

With the exception of Mutual Information (Abbasi-Yadkori et al., 2024) and Sequence-Probability (Aichberger et al., 2024), which provide theoretical justification, nearly all of these approaches rely on heuristics. Furthermore, none of them have been specifically designed or evaluated for contextual QA.

Beyond single-model uncertainty estimation, several works propose Bayesian frameworks to decompose epistemic and aleatoric uncertainty (Ling et al., 2024; Hou et al., 2024). Ling et al. (2024) use a Bayesian in-context learning formulation, quantifying epistemic uncertainty as the mutual information between the model's output and the in-context examples. Hou et al. (2024) introduce a generic Bayesian framework based on input clarification, where epistemic uncertainty is again defined as mutual information between outputs under different clarified queries. Although theoretically grounded, these methods require replacing the LLM with a Bayesian framework and computing uncertainty over the Bayesian model, which is computationally expensive. In contrast, we quantify the uncertainty of a single LLM directly.

Only a little number of of recent works have directly addressed UQ in contextual QA or retrieval-augmented generation (RAG). Soudani et al. (2025) propose an axiomatic framework for diagnosing deficiencies in existing methods and present a generic UQ method that can be layered on top of other approaches. Perez-Beltrachini & Lapata (2025) introduce a passage-utility based metric, training a lightweight neural model to predict the usefulness of retrieved passages for a given QA task. Similarly, Fadeeva et al. (2025) propose a method that evaluates factuality by jointly assessing faithfulness and factual correctness under both faithful and unfaithful retrieval conditions.

However, all of these methods remain heuristic and largely empirical. In contrast, our work introduces a UQ approach with a grounded theoretical formulation, designed specifically to contextual QA.

### A.4 EXPERIMENTAL DETAILS

#### A.4.1 PROMPTS USED IN THE EXPERIMENTS

To query the model in the contextual QA setting, we use the following prompt:

```
Context: {context}. Here is a given context. You are a helpful assistant.
Answer the following question with a brief but complete response.
Use the context information to answer this question.

Question: {question}
Answer:
```

For the LLM-judge, which evaluates the correctness of generated answers, we use the following prompt:

```
You will be given a context, a question relevant to that context, a predicted
answer, and a list of possible ground truth answers provided by human experts.
```

```
Your task is to assign one of the following labels:
["CORRECT", "INCORRECT", "NOT_ATTEMPTED"].

Context: {context}
Question: {question}
Possible Ground Truth Answers Provided by Human Experts: {ground_truths}
Predicted Answer: {answer}

Labeling instructions:
- Assign "NOT_ATTEMPTED" if the predicted answer fails to engage with the
  question, or explicitly states that an answer is not found in the context.
- Assign "CORRECT" if the predicted answer is accurate based on the context,
  even if it is not explicitly listed among the ground truth answers.
- Assign "INCORRECT" if the predicted answer is contradicted by the context
  or otherwise wrong.

Important: The ground truth list may not cover all valid answers.
For those cases, look at the context.

Provide your explanation and then at the end give your grade.

Response:
```

### A.4.2 PROMPTS FOR FEATURE EXTRACTION

**Honesty.**   To extract the honesty feature, we design positive and negative prompts that encourage truthful versus intentionally false responses.

**Positive:**

```
Context: {context}.
Here is a given context. You are a helpful assistant.
Answer the following question with a brief but complete response.
Be honest with your answer. Do not give unfactual information. Do not lie.

Question: {question}
Answer:
```

**Negative:**

```
Context: {context}.
Here is a given context. You are a helpful assistant.
Answer the following question with a brief but complete response.
Be a liar. Give unfactual information. Tell a lie.

Question: {question}
Answer:
```

—

**Context-Reliance.**   To extract the context-reliance feature, we contrast prompts that either require the model to ground its answer in the provided context or explicitly ignore it.

**Positive:**

```
Context: {context}.
Here is a given context. You are a helpful assistant.
Answer the following question with a brief but complete response.
Use the context information to answer this question.
Do not use your own knowledge. Just look at the context.
```

| Model | Method | Narrative | Qasper | Hotpot |
|---|---|---|---|---|
| **Qwen2.5 - 7B** – PRR | Saplma | 51.7 | 61.6 | 56.2 |
| | **Feature-Gaps** | **57.8** | 58.1 | 65.6 |
| | **Feature-Gaps (10 layers)** | 57.4 | **65.2** | **66.2** |
| **LLama3.1 - 8B** – PRR | Saplma | 62.3 | 60.6 | 60.9 |
| | **Feature-Gaps** | 60.7 | 65.7 | 66.4 |
| | **Feature-Gaps (10 layers)** | **64.3** | **69.3** | **70.4** |
| **Mistralv0.3** - 7B – PRR | Saplma | **56.2** | 43.7 | 51.8 |
| | **Feature-Gaps** | 43.1 | 59.1 | 53.6 |
| | **Feature-Gaps (10 layers)** | 47.2 | **59.0** | **60.7** |
| **Qwen2.5 - 7B** – AUROC | Saplma | 72.7 | 74.9 | 73.8 |
| | **Feature-Gaps** | 73.1 | 73.1 | 78.1 |
| | **Feature-Gaps (10 layers)** | **74.2** | **76.5** | **78.2** |
| **LLama3.1 - 8B** – AUROC | Saplma | 75.1 | 76.9 | 76.4 |
| | **Feature-Gaps** | 73.9 | 75.9 | 77.5 |
| | **Feature-Gaps (10 layers)** | **76.5** | **78.3** | **79.8** |
| **Mistralv0.3** – AUROC | Saplma | **74.2** | 68.2 | 71.1 |
| | **Feature-Gaps** | 68.3 | **75.0** | 70.8 |
| | **Feature-Gaps (10 layers)** | 69.9 | 74.8 | **75.7** |

Table 5: PRR and AUROC metrics for Qwen, LLaMA, and Mistral with best-performing scores in bold.

```
Question: {question}
Answer:
```

**Negative:**

```
Context: {context}.
Here is a given context. You are a helpful assistant.
Answer the following question with a brief but complete response.
DO NOT use the context information to answer this question.
Use your own knowledge. Ignore the context.

Question: {question}
Answer:
```

—

**Context Comprehension.** For context comprehension, we use the regular contextual QA prompt but append the ground-truth answer to the context, simulating an idealized model where the model has already extracted the necessary information.

## A.5 ADDITIONAL EXPERIMENTS

In this section, we provide our additional results.

### A.5.1 SCALING THE NUMBER OF TRAINING SAMPLES

In addition to low-data regime experiments, we also try to scale our method to more training data. We add a small modification to our framework by instead of selecting a single direction $v_i$ for each feature, we extend the approach to select (N) directions from multiple layers. We scaled the training data to 1000 samples for each dataset, and for the scalable version we selected 10 layers per feature. The results are in Table A.5.1

As the results indicate, our method does not benefit substantially from simply increasing the number of training samples, but selecting multiple layers significantly improves performance. By con-

trast, SAPLMA also does not improve much with additional data, and our method almost remains consistently superior.

We also performed similar experiments with more data (5000 samples) on HotPotQA. We could not scale NarrativeQA because we were unable to find a 5000-sample subset whose contexts fit within GPU memory, and Qasper does not contain 5000 samples. The HotPotQA results with 5000 samples are included below.

The results are in A.5.1, and we observe a performance drop for SAPLMA compared to the 1000-sample setting. This is expected: the 5000-sample experiment uses data from the training split of HotPotQA, whereas the earlier 1000-sample experiment used a split from the validation set. This distribution shift appears to affect SAPLMA more strongly than our method. Scaled version of our method is superior to both SaPLMA and our method.

| Method | Model | AUROC | PRR |
|---|---|---|---|
| Feature-Gaps | Qwen2.5 - 7B | 69.8 | 50.9 |
| | LLama3.1 - 8B | 77.5 | 66.3 |
| | Mistralv0.3 - 7B | 71.3 | 54.4 |
| Feature-Gaps (10 Layers) | Qwen2.5 - 7B | 71.6 | 55.9 |
| | LLama3.1 - 8B | 79.4 | 68.9 |
| | Mistralv0.3 - 7B | 71.3 | 54.2 |
| Saplma | Qwen2.5 - 7B | 61.8 | 30.2 |
| | LLama3.1 - 8B | 73.0 | 55.9 |
| | Mistralv0.3 - 7B | 71.1 | 52.8 |

Table 6: HotpotQA Results with 5000 Training Samples

### A.5.2    EXPERIMENTS WITH BIGGER MODELS (QWEN32B)

We evaluated our method on a larger model, Qwen2.5-32B. Due to computational constraints, we were unable to run experiments on Qasper and NarrativeQA, as their long contexts caused GPU memory errors with this model. Therefore, we report only the HotPotQA results in Table A.5.2.

The results show that our method remains noticeably superior to SAPLMA. However, Perplexity and PTrue achieve the strongest performance among all baselines. Since both rely on model probability estimates, this suggests that larger models may produce more calibrated probability signals compared to smaller models, which could explain their stronger performance in this setting.

### A.5.3    SIGNIFICANCE TESTS ON OOD EXPERIMENTS

We also report AUROC scores for the OOD experiments along with statistical significance tests using the DeLong method DeLong et al. (1988). As shown in Table A.5.3, our method remains mostly superior. In all cases where the AUROC difference is substantial, the improvements are statistically significant (p-value $< 0.05$).

| Method | AUROC | PRR |
|---|---|---|
| SemanticEntropy | 66.2 | 42.1 |
| Confidence | 75.4 | 58.5 |
| Entropy | 64.3 | 39.6 |
| EccentricityUncertainty | 66.7 | 36.1 |
| KernelLanguageEntropy | 65.9 | 43.6 |
| ContextCheck | 50.4 | 13.2 |
| PTrue | 78.9 | 63.9 |
| MARS | 71.2 | 51.1 |
| MiniCheckMethod | 71.7 | 48.3 |
| SAR | 67.8 | 44.1 |
| MatrixDegreeUncertainty | 64.6 | 31.7 |
| SumEigenUncertainty | 64.6 | 31.7 |
| Saplma | 58.1 | 32.5 |
| FeatureGaps | 67.5 | 49.0 |

Table 7: HotpotQA, AUORC and PRR Scores with Qwen2.5 - 32B Model

| Model | Training Dataset / Test Dataset | Qasper | HotpotQA | NarrativeQA |
|---|---|---|---|---|
| LLama3.1 - 8B – Feature Gaps | Qasper | 75.3 | 76.9 | 76.3 |
| | HotpotQA | 75.4 | 78.0 | 79.8 |
| | NarrativeQA | 72.4 | 75.8 | 74.8 |
| LLama3.1 - 8B – SAPLMA | Qasper | 74.7 | 69.6 | 72.2 |
| | HotpotQA | 64.8 | 72.8 | 63.4 |
| | NarrativeQA | 68.4 | 71.0 | 67.6 |
| LLama3.1 - 8B – p-values | Qasper | 0.8095 | 4.491e-06 | 5.163e-02 |
| | HotpotQA | 4.096e-04 | 6.314e-04 | 2.861e-13 |
| | NarrQA | 0.1531 | 3.668e-03 | 1.391e-03 |
| Mistralv0.3 - 7B – Feature Gaps | Qasper | 76.0 | 69.2 | 65.2 |
| | HotpotQA | 76.0 | 71.5 | 66.2 |
| | NarrativeQA | 73.0 | 68.9 | 65.1 |
| Mistralv0.3 - 7B – SAPLMA | Qasper | 69.1 | 64.1 | 55.9 |
| | HotpotQA | 64.8 | 73.3 | 66.3 |
| | NarrativeQA | 67.7 | 66.2 | 71.3 |
| Mistralv0.3 - 7B – p-values | Qasper | 3.255e-03 | 2.576e-03 | 1.100e-04 |
| | HotpotQA | 1.100e-05 | 2.744e-01 | 9.629e-01 |
| | NarrQA | 4.349e-02 | 1.744e-01 | 9.634e-03 |
| Qwen2.5 - 7B – Feature Gaps | Qasper | 72.7 | 67.5 | 63.1 |
| | HotpotQA | 64.6 | 76.0 | 73.4 |
| | NarrativeQA | 67.7 | 66.0 | 70.7 |
| Qwen2.5 - 7B – SAPLMA | Qasper | 75.2 | 67.5 | 71.8 |
| | HotpotQA | 61.0 | 76.2 | 63.9 |
| | NarrativeQA | 71.0 | 61.5 | 68.8 |
| Qwen2.5 - 7B – p-values | Qasper | 3.125e-01 | 9.711e-01 | 2.804e-04 |
| | HotpotQA | 2.703e-01 | 9.295e-01 | 4.940e-04 |
| | NarrQA | 2.104e-01 | 3.827e-02 | 4.093e-01 |

Table 8: OOD experiments, AUROC scores with significance values

