# OpenReview forum: "Uncertainty as Feature Gaps: Epistemic Uncertainty Quantification of LLMs in Contextual Question-Answering"
_ICLR.cc/2026/Conference — ICLR 2026 Poster_

### Official Review · Reviewer_WhVN · 2025-10-31

**Soundness:** 3
**Presentation:** 4
**Contribution:** 4
**Rating:** 8
**Confidence:** 3

**Summary:**

The authors propose an uncertainty quantification method derived from an approximation of the upper bound for epistemic uncertainty.  They comprehensively justify the theoretical basis for this method and demonstrate that it empirically works effectively with three different models on three datasets.

**Strengths:**

1. Strong theoretical justification for the validity of this uncertainty quantification method, which is uncommon for many uncertainty quantification methods
2. Strong experimental results, particularly when compared to the computational complexity of baselines
3. Generally, many UQ methods quantify total uncertainty, rather than epistemic uncertainty; this broadens potential applications for this UQ method

**Weaknesses:**

1. Some assumptions are not well-justified, in particular that there exists an optimal token string that would result in an ideal model. This is more or less solely supported by previous work that suggests appending a prompt can be considered to be fine-tuning. While the empirical results suggest this assumption holds true enough for this method to work, I feel that stating that this is a close approximation to a fine-tuned model (lines 179-180) or that it is the “ideal” model (line 190) may need more evidence or more qualifiers.
2. There may be more to add for related work, specifically in the domain of uncertainty decomposition (for instance, https://arxiv.org/abs/2311.08718 and https://arxiv.org/pdf/2402.10189)—I believe this would provide better context for this work.

**Questions:**

Is there a way to organize Table 1 so it’s easier to tell which baselines require multiple samples and which require supervised data? This would make comparisons easier.

---

> ### Author Response · Authors · 2025-11-20
>
> We sincerely appreciate the reviewer’s thoughtful and constructive feedback. Below, we respond to each point individually:
>
> **Concern - 1**:  Some assumptions are not well-justified, in particular that there exists an optimal token string that would result in an ideal model. This is more or less solely supported by previous work that suggests appending a prompt can be considered to be fine-tuning. While the empirical results suggest this assumption holds true enough for this method to work, I feel that stating that this is a close approximation to a fine-tuned model (lines 179-180) or that it is the “ideal” model (line 190) may need more evidence or more qualifiers.
>
> **Answer**:  This is indeed true, but as the reviewer said, we already state that this is an **approximation** to the ideal model, not the exact ideal model. Although the approximation may not be perfect, we rely on the fact that prompting is Turing-complete, meaning that for any computable function, there exists a Transformer and a corresponding prompt that can implement it, as discussed in the paper. Making such assumptions or approximations is also a common starting point in the UQ literature [1 ,2 ,3].
>
> **Suggestion - 1**: There may be more to add for related work, specifically in the domain of uncertainty decomposition (for instance, https://arxiv.org/abs/2311.08718 and https://arxiv.org/pdf/2402.10189) .I believe this would provide better context for this work.
>
> **Answer**: We thank the reviewer for pointing out these papers. We discussed these works and updated our related work section in the revised manuscript.
>
> **Suggestion - 2**:  Is there a way to organize Table 1 so it’s easier to tell which baselines require multiple samples and which require supervised data? This would make comparisons easier.
>
> **Answer**: Thanks for the suggestion! We modified Table 1 in the revised manuscript by categorizing different UQ methods.
>
>
> [1] Kajetan Schweighofer, Lukas Aichberger, Mykyta Ielanskyi, and Sepp Hochreiter. On information- theoretic measures of predictive uncertainty. arXiv preprint arXiv:2410.10786, 2024.
>
> [2] Yasin Abbasi-Yadkori, Ilja Kuzborskij, András György, and Csaba Szepesvari. To believe or not
> to believe your LLM: Iterativeprompting for estimating epistemic uncertainty. In The Thirty-
> eighth Annual Conference on Neural Information Processing Systems, 2024. URL https:
> //openreview.net/forum?id=k6iyUfwdI9.
>
> [3] Nikita Kotelevskii, Vladimir Kondratyev, Martin Takáˇc, Eric Moulines, and Maxim Panov. From
> risk to uncertainty: Generating predictive uncertainty measures via bayesian estimation. In
> The Thirteenth International Conference on Learning Representations, 2025. URL https:
> //openreview.net/forum?id=cWfpt2t37q.

---

> > ### Comment · Reviewer_WhVN · 2025-11-24
> >
> > Thank you for your thoughtful response, which addresses most of my concerns. I believe this is a strong paper and will maintain my score.

---

> > > ### Author Response · Authors · 2025-11-25
> > >
> > > We thank the reviewer for their feedback and supportive comments!

---

### Official Review · Reviewer_id5u · 2025-10-31

**Soundness:** 3
**Presentation:** 3
**Contribution:** 3
**Rating:** 4
**Confidence:** 3

**Summary:**

This paper proposes an information-theoretic metric to quantify epistemic uncertainty, i.e., cross-entropy between the true (ideal) predictive distribution and the given model's distribution --adapted from [Schweighofer et al. (2024)]. It operationalizes it via interpretability tools. Starting from the information-theoretic formulation, the authors derive an upper-bound of the epistemic uncertainty formulated as the difference between the ideal and given model's internal states. These can be represented as decomposed vectors (linear representation hypothesis); furthermore, the ideal and given model's coefficients in the linear representation can be interpreted as the deviation of the ideal from the given model (namely Feature-Gaps). In practice, for contextual QA, three features are selected: Context reliance, Context comprehension, and Honesty. Vectors for these features are extracted using interpretability techniques.

The proposed method based on interpretability tools is sensible, efficient, and shows competitive performance. However details about the feature extraction/approximation are missing and there are some weak points in the experimental setup (see below).

**Strengths:**

- Using interpretability insights to estimate uncertainty is novel.
- The proposed approach is efficient at test time; although it requires training it necessitates only few labelled data points. It does require tuning some hyper-parameters (e.g., layers to use), features, prompts.

**Weaknesses:**

- Missing experiments on RAG contexts, larger models, and p(true) approach. Significance tests on results as sometimes these are quite close values.
- Lack of details about the training of the $w_i$ weights.

**Questions:**

- Experimental setup:

	- The authors should clarify what sort of input is used for HotPotQA, it is common practice for this dataset to be considered an open-book QA task where external retrieval systems are used to retrieve passages for context (RAG). If this is not the case, and passages available in the dataset are used, then all evaluation tasks are instances of Reading Comprehension (RC). In this case the evaluation does not cover the case of real RAG scenarios. How would the approach work on the case of noisy and misleading real retrieval?

	- The evaluation focuses on 3 models of similar size (different families). It would also make a clearer empirical statement of the method's performance if the authors evaluate on larger models, e.g., Qwen2.5-32B.

	- Out-of-distribution experiments (Section 5.3, Figure 3) need to incorporate a non-supervised baseline, e.g., the cheaper unsupervised Perplexity, in order to better visualize how the compared out-of-distribution methods (Feature-Gaps and SAPLMA) compare to this simple approach. Also, by looking at values in Figure 3, we can see that these are PRR (better to add this in the caption), the authors should also report the AUROCs.

	- Significance values should be reported for results on Table 1.

	- An strong widely used approach p(true) is mentioned in the paper though not included in experiments.

- "Feature Extraction" in Section 4.2 could be better explained (add missing details). Line 281,  "access to a set of T labelled samples" I guess this is with gold answer. Line 282, "standard instruction", what would be standard instruction in the case "look at the context" and "se your own knowledge"? How is $\alpha_i$ in Eq. 7 (and Line 304) computed? Is $\alpha_i v_i$ a decomposition of $v^l$? Is in the end $\alpha_i$ assumed to be 1 in Eq. 7? More details about the training of the procedure to obtain the $w_i$ weights are needed (line 305). What is the connection of the $w_i$ weights and the ideal model representations ($w_i$'s are trained to predict correct/incorrect)? What is the relation of training these weights and the ranking formulation in Line 134?

- Related work. The method by [Laura Perez-Beltrachini and Mirella Lapata] trains the so called passage-utility that is predicting correct/incorrect answer. How does this differentiates from the training $w_i$s on the task of predicting correct/incorrect answer? Beyond the different amount of training data required.

---

> ### Author Response · Authors · 2025-11-20
>
> We sincerely appreciate the reviewer’s thoughtful and constructive feedback. Below, we respond to each point individually:
>
> **Concern - 1**: Missing baseline PTrue.
>
> **Answer**: We included PTrue as a baseline and updated Table 1 in the revised manuscript. Below, we share the PTrue results:
>
>
>
> ## Llama8b:
> | Dataset      | AUROC (%) | PRR (%) |
> |--------------|-----------|----------|
> | HotpotQA     | 68.8      | 46.1     |
> | Qasper       | 74.2      | 57.4     |
> | NarrativeQA  | 68.2      | 36.7     |
>
> ## Mistral7b:
> | Dataset      | AUROC (%) | PRR (%)  |
> |--------------|-----------|-----------|
> | HotpotQA     | 49.3      | -9.68     |
> | Qasper       | 36.7      | -51.9     |
> | NarrativeQA  | 56.8      | 9.71      |
>
> ## Qwen7b:
> | Dataset      | AUROC (%) | PRR (%) |
> |--------------|-----------|----------|
> | HotpotQA     | 49.3      | -9.68    |
> | Qasper       | 63.9      | 37.4     |
> | NarrativeQA  | 54.1      | 7.79     |
>
> Compared to our method or even other unsupervised methods, Ptrue couldn't show strong performance in the contextual QA setting in 7/8 B models size.
>
> **Concern - 2**: AUROC results of OOD experiments and significance values
>
> **Answer**:  We also report AUROC scores for the OOD experiments along with statistical significance tests using the DeLong method [1]. As shown in the Table below, our method remains mostly superior. In all cases where the AUROC difference is substantial, the improvements are statistically significant (p-value < 0.05).
>
> # LLAMA
>
> ## LLAMA — Feature Gaps
> | Train \ Test | Qasper | HotpotQA | NarrativeQA |
> |--------------|--------|----------|-------------|
> | Qasper       | 75.3   | 76.9     | 76.3        |
> | HotpotQA     | 75.4   | 78.0     | 79.8        |
> | NarrativeQA  | 72.4   | 75.8     | 74.8        |
>
> ## LLAMA — SAPLMA
> | Train \ Test | Qasper | HotpotQA | NarrativeQA |
> |--------------|--------|----------|-------------|
> | Qasper       | 74.7   | 69.6     | 72.2        |
> | HotpotQA     | 64.8   | 72.8     | 63.4        |
> | NarrativeQA  | 68.4   | 71.0     | 67.6        |
>
> ## LLAMA — p-values
> | Train \ Test | Qasper   | HotpotQA  | NarrativeQA |
> |--------------|----------|-----------|-------------|
> | Qasper       | 0.8095   | 4.491e-06 | 5.163e-02   |
> | HotpotQA     | 4.096e-04| 6.314e-04 | 2.861e-13   |
> | NarrativeQA  | 0.1531   | 3.668e-03 | 1.391e-03   |
>
>
> # Mistral
>
> ## Mistral — Feature Gaps
> | Train \ Test | Qasper | HotpotQA | NarrativeQA |
> |--------------|--------|----------|-------------|
> | Qasper       | 76.0   | 69.2     | 65.2        |
> | HotpotQA     | 76.0   | 71.5     | 66.2        |
> | NarrativeQA  | 73.0   | 68.9     | 65.1        |
>
> ## Mistral — SAPLMA
> | Train \ Test | Qasper | HotpotQA | NarrativeQA |
> |--------------|--------|----------|-------------|
> | Qasper       | 69.1   | 64.1     | 55.9        |
> | HotpotQA     | 64.8   | 73.3     | 66.3        |
> | NarrativeQA  | 67.7   | 66.2     | 71.3        |
>
> ## Mistral — p-values
> | Train \ Test | Qasper   | HotpotQA | NarrativeQA |
> |--------------|----------|----------|-------------|
> | Qasper       | 3.255e-03| 2.576e-03| 1.100e-04   |
> | HotpotQA     | 1.100e-05| 2.744e-01| 9.629e-01   |
> | NarrativeQA  | 4.349e-02| 1.744e-01| 9.634e-03   |
>
>
> # Qwen8b
>
> ## Qwen8b — Feature Gaps
> | Train \ Test | Qasper | HotpotQA | NarrativeQA |
> |--------------|--------|----------|-------------|
> | Qasper       | 72.7   | 67.5     | 63.1        |
> | HotpotQA     | 64.6   | 76.0     | 73.4        |
> | NarrativeQA  | 67.7   | 66.0     | 70.7        |
>
> ## Qwen8b — SAPLMA
> | Train \ Test | Qasper | HotpotQA | NarrativeQA |
> |--------------|--------|----------|-------------|
> | Qasper       | 75.2   | 67.5     | 71.8        |
> | HotpotQA     | 61.0   | 76.2     | 63.9        |
> | NarrativeQA  | 71.0   | 61.5     | 68.8        |
>
> ## Qwen8b — p-values
> | Train \ Test | Qasper   | HotpotQA | NarrativeQA |
> |--------------|----------|----------|-------------|
> | Qasper       | 3.125e-01| 9.711e-01| 2.804e-04   |
> | HotpotQA     | 2.703e-01| 9.295e-01| 4.940e-04   |
> | NarrativeQA  | 2.104e-01| 3.827e-02| 4.093e-01   |
>
>
> [1] Elizabeth R. DeLong, David M. DeLong, and Daniel L. Clarke-Pearson. Comparing the areas
> under two or more correlated receiver operating characteristic curves: A nonparametric approach.
> Biometrics, 44(3):837–845, 1988. ISSN 0006341X, 15410420. URL http://www.jstor.
> org/stable/2531595

---

> ### Author Response · Authors · 2025-11-20
>
> **Concern - 3**: The authors should clarify what sort of input is used for HotPotQA, it is common practice for this dataset to be considered an open-book QA task where external retrieval systems are used to retrieve passages for context (RAG). If this is not the case, and passages available in the dataset are used, then all evaluation tasks are instances of Reading Comprehension (RC). In this case the evaluation does not cover the case of real RAG scenarios. How would the approach work on the case of noisy and misleading real retrieval?
>
> **Answer**: We use the passages provided in the HotPotQA dataset as context. For this reason, we frame our setting as contextual question answering rather than RAG. The randomness introduced by external retrieval systems in real RAG pipelines is indeed an important problem, but it is orthogonal to our contribution. Our method models uncertainty conditioned on **a given/retrieved input**, and focuses on isolating the epistemic component. In principle, retrieval uncertainty can be additively included in our formulation, but doing so would require explicitly modeling the retriever’s randomness, which is outside the scope of this work. We have now added a short discussion of this limitation and its relevance to future work in the updated version of the manuscript
>
> **Concern - 4**:  "Feature Extraction" in Section 4.2 could be better explained (add missing details)
>
> **Answer**: Let us clarify the components of the feature-extraction procedure and the role of the parameters in Eq. 7.
>
> During training, the “T labelled samples” refer to question–context pairs together with their gold (short) answers, as the reviewer inferred. Using these samples, we extract the three feature vectors $v_i$ following the procedure in Section 4.2 (feature extraction part).
>
> At test time, all methods, including ours, use the same standard instruction, which simply asks the model to answer the question given the provided context (prompt in Appendix A.4.1).
>
> At test time, given the model’s generated answer, we obtain the hidden states at each token position. We then compute the dot product between these hidden states and each feature vector (v_i), which gives the activation coefficients $\alpha_i$. These $\alpha_i$ measure how strongly the model’s response aligns with each extracted feature direction.
>
> The parameters $\beta_i$ are not directly observable: they correspond to the contribution that an ideal model would assign to each direction. Therefore, we estimate them during training. Since epistemic uncertainty should correlate with correctness, we learn $\beta_i$ by minimizing the cross-entropy between the correctness labels and the uncertainty score (Eq. 7, left side).
>
> To keep the number of trainable parameters small, we parameterize $\beta_i$ as a scaled version of $\alpha_i$, i.e., $\beta_i = w_i \alpha_i$. This leads to the form in the right side of Eq. 7. Other parameterizations (including nonlinear combinations of the $\alpha_i$ are possible, but we choose this simple form for stability and interpretability.
>
> This training objective is consistent with the ranking-based motivation described earlier (Line 134), since learning the weights $w_i$ encourages higher uncertainty scores for incorrect answers and lower ones for correct answers by cross-entropy loss, which would lead to better ranking.
>
> We hope this would clarify our algorithmic approach and its connection to our theory.
>
>
> **Concern - 5**: Out-of-distribution experiments (Section 5.3, Figure 3) need to incorporate a non-supervised baseline, e.g., the cheaper unsupervised Perplexity.
>
> **Answer**: We did not include the unsupervised methods in the OOD experiments because their performance would be identical to what is already reported in Table 1 (as there is no training). When compared to these unsupervised baselines, our method remains mostly superior even in the worst-case OOD setting.
>
> **Concern - 6**: Comparison with [Laura Perez-Beltrachini and Mirella Lapata] related work
>
> **Answer**: Our training procedure serves a fundamentally different purpose from that of Perez-Beltrachini & Lapata. We train only a small number of parameters to estimate the $\beta_i$​ values, using the feature magnitudes $\alpha_i$​ derived from the model’s hidden states for a given question–context pair. This allows us to quantify the model’s epistemic uncertainty conditioned on the provided context. In contrast, Perez-Beltrachini & Lapata train a quality estimator that takes the entire retrieved passage as input (as text) and predicts its utility for answering the question. Thus, their method focuses on evaluating context quality, whereas our method focuses on evaluating the model’s epistemic uncertainty given a fixed context. The approaches are therefore largely orthogonal, beyond differences in required training data.

---

> ### Author Response · Authors · 2025-11-20
>
> **Concern - 7**: The evaluation focuses on 3 models of similar size (different families). It would also make a clearer empirical statement of the method's performance if the authors evaluate on larger models, e.g., Qwen2.5-32B.
>
> **Answer**: The main reason we did not evaluate larger models is our computational constraints. With 8×A100 (40GB) GPUs, we were unable to run experiments on datasets with long contexts such as Qasper and NarrativeQA. However, following the reviewer’s suggestion, we were able to run experiments on HotPotQA with Qwen2.5-32B, which has shorter contexts and fits within our resources. The results are provided below and have been added to the revised manuscript.
>
> | Method                           | AUROC | PRR  |
> |----------------------------------|-------|------|
> | SemanticEntropy                  | 66.2  | 42.1 |
> | Perplexity                     | 75.4  | 58.5 |
> | Entropy                          | 64.3  | 39.6 |
> | Eccentricity         | 66.7  | 36.1 |
> | KernelLanguageEntropy            | 65.9  | 43.6 |
> | ContextCheck                     | 50.4  | 13.2 |
> | PTrue                            | 78.9  | 63.9 |
> | MARS                             | 71.2  | 51.1 |
> | MiniCheck                 | 71.7  | 48.3 |
> | SAR                              | 67.8  | 44.1 |
> | Saplma                           | 58.1  | 32.5 |
> | FeatureGaps                      | 67.5  | 49.0 |
>
> The results show that our method remains noticeably superior to SAPLMA. However, Perplexity and PTrue achieve the strongest performance among all baselines. Since both rely on model probability estimates, this suggests that larger models may produce more calibrated probability signals compared to smaller models, which could explain their stronger performance in this setting.

---

### Official Review · Reviewer_Eyf6 · 2025-11-03

**Soundness:** 3
**Presentation:** 3
**Contribution:** 4
**Rating:** 6
**Confidence:** 5

**Summary:**

The paper presents a new supervised method for uncertainty quantification for LLMs. The features are constructed as the difference between hidden states of LLM prompted in the standard way and the LLM which was prompted to be “liar” or “do not look at the context”.
The authors claim that constructing features in this way results in better generalization than standard hidden states or features built from attention maps (such as Lookback lens).
The model is basically a linear regression on top of the hidden states.
Important contribution of the paper is that the authors show that the epistemic uncertainty can be bounded from above by such a linear expression.

One of the main problems of the paper is that the authors claim that they quantify only epistemic uncertainty. However, the tasks where they test their methods assume that both types of uncertainty would be useful, so it seems that there is no need for such disentanglement.

Another problem is that the method needs sampling of the answer multiple times, which introduces large computational overhead (basically up to 3-4 times). These is very close to sampling-based methods that need as little as 5 samples to show improvements over the simple baselines. Sampling alone should carry some additional information that help to improve the performance of hallucination detection. So, the substantial part of the improvement might come from sampling rather than other parts of the method.

The set of baselines is good, however, there are few methods that explore the idea of the difference between “good” and “bad” answers of LLM.
Vazhentsev, Artem, et al. "Token-Level Density-Based Uncertainty Quantification Methods for Eliciting Truthfulness of Large Language Models." Proceedings of the 2025 Conference of the Nations of the Americas Chapter of the Association for Computational Linguistics: Human Language Technologies (Volume 1: Long Papers). 2025.
Comparison to this baseline looks important for me

The authors claim that the method works well with small amount of training data, which is good. However, the analysis in the case when the amount of data is bigger is not conducted. It would interesting to see wheather there are improvements when the number of train instances is relatively large 5000+.
Note that the lack of the improvements doex not deteriorate the method, but the lack of such analysis is a notable drawback of this work. This analysis is important for me.

Please, also note that in the experiment with generalization you compare your method with SAPLMA. However, the authors of Lookback lens showed that SAMPLA in general is not very good at generalization. It would be reasonable to compare your method to Lookback lens, as the main point of their paper was better generalization than SAPLMA. This analysis is moderately important, but nice to have.

Overall, I think the work is good and open for the discussion.

**Strengths:**

1.	Interesting method that has better performance than other supervised baselines.
2.	Important contribution of the paper is that the authors show that the epistemic uncertainty can be bounded from above by such a linear expression.

**Weaknesses:**

1.	Need additional analysis with more training data
2.	Comparison to another baseline
3.	The method is computationally expensive, because it needs to sample 3-4 times
4.	The improvements might come not from the “epistemic nature” of uncertainty, but just because of sampling.
5.	The authors show that benefits of their method that disentangles epistemic uncertainty on tasks that need both epistemic and aleatoric uncertainty.

**Questions:**

1.	Interesting method that has better performance than other supervised baselines.
2.	Important contribution of the paper is that the authors show that the epistemic uncertainty can be bounded from above by such a linear expression.

---

> ### Author Response · Authors · 2025-11-20
>
> We sincerely appreciate the reviewer’s thoughtful and constructive feedback. Below, we respond to each point individually:
>
> **Concern - 1**: One of the main problems of the paper is that the authors claim that they quantify only epistemic uncertainty. However, the tasks where they test their methods assume that both types of uncertainty would be useful, so it seems that there is no need for such disentanglement.
>
> **Answer**: Our task is to detect when the model makes mistakes or hallucinates in contextual QA. Aleatoric uncertainty reflects inherent ambiguity that belongs to the task/data, for example, producing paraphrased but equivalent answers (“Paris” vs. “It’s Paris”) corresponds to the aleatoric uncertainty. In our setting, this form of uncertainty is not problematic. since these cases are not errors. What matters for hallucination detection is **epistemic uncertainty**: uncertainty stemming from the model’s lack of knowledge or ability to answer correctly. This distinction is well established in the UQ literature, including recent work on LLMs [1 ,2] . If one interprets aleatoric uncertainty as randomness from retrieval in RAG systems, that would need to be explicitly modeled. In our setting, however, the input x (including context) is assumed already been retrieved/given. We intentionally avoid the term “RAG” because we do not model retrieval randomness; our analysis focuses solely on the model’s uncertainty conditioned on a provided context. Handling uncertainty arising from retrieval is indeed important future work and orthogonal to our work, but it is outside the scope of our current formulation.
>
> **Concern - 2**: Another problem is that the method needs sampling of the answer multiple times, which introduces large computational overhead (basically up to 3-4 times).
>
> **Answer**: We believe there is a misunderstanding of our method. We **do not perform sampling** to compute the uncertainty score. At test time, the model generates a single answer, and we simply extract the hidden states of its tokens at selected layers. The uncertainty score is obtained by taking dot products between these hidden states and the three feature vectors extracted previously. The only stage that requires multiple forward passes is the **feature extraction step during training**, as shown in Figure 2. This step is performed once to obtain the three feature vectors. After that, test-time inference requires **no sampling and only one forward pass**, that’s why we claim our method is computationally efficient.
>
> **Concern-3**: Additional Baseline: Token-Level Density-Based Uncertainty Quantification Methods for Eliciting Truthfulness of Large Language Models.
>
> **Answer**: We thank the reviewer for pointing out this work. Following the reviewer’s suggestion, we implemented the proposed method, Average Token-level Mahalanobis Distances (ATMD), in our benchmark, and we report the results below.
>
> ## Llama
> | Dataset     | AUROC | PRR  |
> |-------------|--------|------|
> | Qasper      | 62.8   | 32.0 |
> | HotpotQA    | 60.9   | 26.8 |
> | NarrativeQA | 57.1   | 21.1 |
>
> ## Qwen
> | Dataset     | AUROC | PRR  |
> |-------------|--------|------|
> | Qasper      | 66.1   | 42.4 |
> | HotpotQA    | 58.8   | 22.3 |
> | NarrativeQA | 57.9   | 19.4 |
>
> ## Mistral
> | Dataset     | AUROC | PRR  |
> |-------------|--------|------|
> | Qasper      | 66.4   | 37.7 |
> | HotpotQA    | 68.1   | 43.4 |
> | NarrativeQA | 59.5   | 21.6 |
>
> In our experiments, the performance of ATMD is not even close to SAPLMA when using 256 samples and is far below our method. We split these 256 samples into 128 for extracting the distances and 128 for training the classifier. We believe this performance gap is reasonable: as shown in their paper (Table 1), most of the gains over SAPLMA come from combining ATMD with additional uncertainty scores, such as sequence probability. That's why, without combining with sequence probability, its raw performance is low compared to other supervised methods.
>
> We have added these new results to the updated Table 1 in the revised manuscript.
>
> **Concern - 4**: Missing OOD Experiments of Lookbacklens
>
> **Answer**: We agree with the reviewer on this point. However, the main reason we could not conduct this experiment is that LookbackLens requires attention maps for every layer, which leads to out-of-memory errors on our hardware (8×A100 40GB) when using the HuggingFace implementation. For this reason, we were only able to run LookbackLens experiments on HotPotQA and could not perform the OOD experiments.
>
>
>
> [1] Yasin Abbasi-Yadkori, Ilja Kuzborskij, András György, and Csaba Szepesvari. To believe or not
> to believe your LLM: Iterative prompting for estimating epistemic uncertainty. Neurips 2024
>
> [2] Nikita Kotelevskii, Vladimir Kondratyev, Martin Takáˇc, Eric Moulines, and Maxim Panov. From
> risk to uncertainty: Generating predictive uncertainty measures via bayesian estimation. ICLR 2025

---

> ### Author Response · Authors · 2025-11-20
>
> **Concern - 5**: The authors claim that the method works well with small amount of training data, which is good. However, the analysis in the case when the amount of data is bigger is not conducted.
>
> **Answer**: This is indeed an important point, and we thank the reviewer for highlighting it. This observation also motivated us to explore a more scalable version of the feature-gaps idea. Instead of selecting a single direction $v_i$ for each feature, we extend the approach to select (N) directions from multiple layers. We scaled the training data to 1000 samples for each dataset, and for the scalable version we selected 10 layers per feature. The results are shown below and have been added to the Appendix of the revised manuscript.
>
> ## Qwen – PRR
> | Method | Qasper | Hotpot | Narrative |
> |--------|--------|--------|-----------|
> | **Saplma** | 61.6 | 56.2 | 51.7 |
> | **Feature-Gaps** | 58.1 | 65.6 | 57.8 |
> | **Feature-Gaps (10 layers)** | 65.2 | 66.2 | 57.4 |
>
> ---
>
> ## LLaMA – PRR
> | Method | Qasper | Hotpot | Narrative |
> |--------|--------|--------|-----------|
> | **Saplma** | 60.6 | 60.9 | 62.3 |
> | **Feature-Gaps** | 65.7 | 66.4 | 60.7 |
> | **Feature-Gaps (10 layers)** | 69.3 | 70.4 | 64.3 |
>
> ---
>
> ## Mistral – PRR
> | Method | Qasper | Hotpot | Narrative |
> |--------|--------|--------|-----------|
> | **Saplma** | 43.7 | 51.8 | 56.2 |
> | **Feature-Gaps** | 59.1 | 53.6 | 43.1 |
> | **Feature-Gaps (10 layers)** | 59.0 | 60.7 | 47.2 |
>
>
> As the results indicate, our method does not benefit substantially from simply increasing the number of training samples, but selecting multiple layers significantly improves performance. By contrast, SAPLMA also does not improve much with additional data, and our method remains superior.
>
>
> We also performed similar experiments with more data (5000 samples) on HotPotQA. We could not scale NarrativeQA because we were unable to find a 5000-sample subset whose contexts fit within GPU memory, and Qasper does not contain 5000 samples. The HotPotQA results with 5000 samples are included below.
>
> ## Qwen
> | Method                   | PRR  | AUROC |
> |--------------------------|------|--------|
> | Feature-Gaps             | 50.9 | 69.8   |
> | Feature-Gaps (10 Layers) | 55.9 | 71.6   |
> | Saplma                   | 30.2 | 61.8   |
>
> ## LLama
>
> | Method                   | PRR  | AUROC |
> |--------------------------|------|--------|
> | Feature-Gaps             | 66.3 | 77.5   |
> | Feature-Gaps (10 Layers) | 68.9 | 79.4   |
> | Saplma                   | 55.9 | 73.0   |
>
> ## Mistral
>
> | Method                   | PRR  | AUROC |
> |--------------------------|------|--------|
> | Feature-Gaps             | 54.4 | 71.3   |
> | Feature-Gaps (10 Layers) | 54.2 | 71.3   |
> | Saplma                   | 52.8 | 71.1   |
>
>
>
>
>
> The results are consistent with the 5000-sample setting as well. Increasing the number of layers has a positive effect on our method’s performance, and our method remains superior to SAPLMA. For some models, SAPLMA shows a performance drop compared to the 1000-sample experiments. This is because, in order to scale HotPotQA to 5000 samples, we used the training set, whereas in the previous experiment, the 1000-sample training split came from the validation set. This distribution shift negatively affects SAPLMA.
>
>
> Overall, we believe the feature-gap framework can be made scalable by selecting more features and more layers per feature. We discuss these ideas further in the limitations and future-work section. We again thank the reviewer for these valuable comments.

---

### Author Response · Authors · 2025-12-04
**Overview of the Rebuttal**

As we conclude the rebuttal phase, we would like to summarize our interactions with the reviewers.

**Reviewer Eyf6** found our paper interesting, appreciated the value of our epistemic uncertainty bound, and considered the work worthy of discussion. We addressed their concerns regarding the separation of aleatoric and epistemic uncertainty, clarifying the conceptual foundation using established frameworks. We also clarified the misunderstanding about run-time and sampling, implemented the requested ATMD baseline (showing our method remains superior), and provided data-scaling experiments. We believe these responses resolve all major concerns.


**Reviewer id5u** highlighted that our algorithm is novel, efficient at run time, and data-efficient. We clarified the distinction between our setting and RAG, added the requested PTrue baseline, provided AUROC significance tests, and explained the training of the w_i parameters and their connection to our theoretical formulation. We also clarified the novelty of our approach relative to the work of Perez-Beltrachini & Lapata.


**Reviewer WhVN** emphasized that our method is theoretically grounded (something uncommon in the UQ literature) and appreciated its strong empirical performance and the clear separation of epistemic and aleatoric uncertainty. We addressed their concerns about the assumptions underlying our theoretical results, supported by relevant prior work. As noted in their rebuttal reply, the reviewer confirmed that their concerns were resolved.


Overall, we thank all reviewers for their thoughtful comments and constructive feedback

---

### Meta-Review · Area_Chair_w9Be · 2026-01-06

**Summary:**

This paper introduces a method to quantify the "epistemic uncertainty" (uncertainty arising from a lack of knowledge) in Large Language Models during contextual question-answering. The authors propose that this uncertainty can be measured as a "feature gap" between the model’s current internal states and those of an idealized version of the model. Reviewers praised the paper for its strong theoretical grounding and the efficiency of the proposed method, which performs well with very little training data. Initial concerns centered on the lack of comparison with certain baselines, the need for testing on larger models, and a misunderstanding regarding the computational cost of the method at test time.

**Reviewer Concerns:**

The authors provided a very thorough rebuttal that addressed nearly all of the reviewers' concerns. They clarified that their method is highly efficient at test time and does not require multiple sampling passes, which corrected a major misunderstanding. Furthermore, they added several requested baselines (such as PTrue and ATMD), conducted experiments on a larger 32B model, and provided statistical significance tests. While Reviewer WhVN noted that some theoretical assumptions about the "ideal model" are approximations, the authors’ explanation regarding the flexibility of prompting and the strong empirical results satisfied the reviewers. The distinction between contextual QA and full RAG was also clarified as a matter of scope.

**Reviewer Scores:**

Reviewer WhVN (8) is a strong supporter and maintained their high score after the rebuttal. Reviewer Eyf6 (6) expressed that the work is good and open for discussion; given that their main concern about computational overhead was a misunderstanding cleared up by the authors, they would likely lean toward a solid 7. Reviewer id5u (4) was the most critical but gave the authors a clear list of missing experiments to improve the paper. Since the authors provided almost everything requested (new baselines, larger model results, and significance tests), it is highly probable that this reviewer would move their score to a 6 (marginally above acceptance). Therefore, the consensus points toward acceptance.

---

### Decision · Program_Chairs · 2026-01-26

Accept (Poster)